# N-Terminal Sequences of Signal Peptides Assuming Critical Roles in Expression of Heterologous Proteins in *Bacillus subtilis*

**DOI:** 10.3390/microorganisms12071275

**Published:** 2024-06-23

**Authors:** Meijuan Zhang, Jie Zhen, Jia Teng, Xingya Zhao, Xiaoping Fu, Hui Song, Yeni Zhang, Hongchen Zheng, Wenqin Bai

**Affiliations:** 1College of Life Science and Agriculture Forestry, Qiqihar University, Qiqihar 161006, China; zhangmeijuan_002@163.com; 2Industrial Enzymes National Engineering Research Center, Tianjin Institute of Industrial Biotechnology, Chinese Academy of Sciences, Tianjin 300308, China; zhen_j@tib.cas.cn (J.Z.);; 3National Center of Technology Innovation for Synthetic Biology, Tianjin 300308, China; 4Tianjin Key Laboratory for Industrial Biological Systems and Bioprocessing Engineering, Tianjin Institute of Industrial Biotechnology, Chinese Academy of Sciences, Tianjin 300308, China; 5College of Food Science and Biotechnology, Tianjin Agricultural University, Tianjin 300392, China; zhangyeni@tjau.edu.cn; 6Key Laboratory of Engineering Biology for Low-Carbon Manufacturing, Tianjin Institute of Industrial Biotechnology, Chinese Academy of Sciences, Tianjin 300308, China

**Keywords:** signal peptide, secretion, transcription, extracellular expression, N-terminal amino acid sequence, alkaline pectin lyase

## Abstract

The N-terminal sequences of proteins and their corresponding encoding sequences may play crucial roles in the heterologous expression. In this study, the secretory expression of alkaline pectin lyase APL in *B. subtilis* was investigated to explore the effects of the N-terminal 5–7 amino acid sequences of different signal peptides on the protein expression and secretion. It was identified for the first time that the first five amino acid sequences of the N-terminal of the signal peptide (SP^-^LipA) from *Bacillus subtilis* lipase A play an important role in promoting the expression of APL. Furthermore, it was revealed that SP^-^LipA resulted in higher secretory expression compared to other signal peptides in this study primarily due to its encoding of N-terminal amino acids with relatively higher transcription levels and its efficient secretion capacity. Based on this foundation, the recombinant strain constructed in this work achieved a new record for the highest extracellular yields of APL in *B. subtilis*, reaching 12,295 U/mL, which was 1.9-times higher than that expressed in the recombinant *Escherichia coli* strain previously reported. The novel theories uncovered in this study are expected to play significant roles in enhancing the expression of foreign proteins both inside and outside of cells.

## 1. Introduction

*Bacillus subtilis* has many attributes such as food security, easy genetic manipulation, and efficient protein secretion ability, which make it a popular host for recombinant protein production [1,2]. It also has long been used as a model organism for molecular research, as well as an industrial workhorse for the production of valuable enzymes [3]. Many strategies including host strain modification, promoter and signal peptide optimization, and fermentation optimization were used in the construction of cell factories of *B. subtilis* to improve the yield of specific proteins [4,5,6,7]. Even so, due to the complex processes of heterologous protein expression, the regulation of protein synthesis in cells remains largely unknown. As a result, the majority of heterologous proteins are generated at extremely low quantities or may even fail to be expressed in recombinant *B. subtilis* strains [4,8,9,10].

Generally, except codon optimization, the initial steps to enhance the yield of enzymes are to optimize molecular tools of the expression system, such as promoters and signal peptides (SPs). However, the optimal expression molecular tools for any given secretory protein cannot currently be predicted through experience or in silico so far [4]. N-terminal signal peptides (SPs) are responsible for guiding the target proteins into the general secretory (Sec) pathway or the twin-arginine protein translocation (TAT) pathway in *B. subtilis* [11,12,13]. Despite the sequence variability of signal peptides, universal biophysical features are preserved which are composed of three parts: N-terminal (N-), hydrophobic helical core (H-), and flexible C-terminal (C-) regions [14]. N-regions are ~5 residues long and commonly have positively charged residue(s) that can interact with the SRP, SecB, or the negatively charged periphery of the signal peptide binding groove of SecA [14]. The TAT secretion pathway exhibits two distinct characteristics when compared to the Sec secretion pathway [15,16]. One is that it accepts cargo proteins with an N-terminal signal peptide that carries the canonical twin-arginine motif which is essential for transport. The other is that the TAT pathway only accepts and translocates fully folded cargo proteins across the respective membrane [15,17,18]. Many studies were performed to improve the extracellular expression of specific proteins by screening an efficient signal peptide belonging to an appropriate secretion pathway [19,20,21,22,23]. However, few pieces of research have paid attention to another property of signal peptides. The N-terminal SP sequence is also the key influence factor for the initiation of protein translation and the stability of mRNA so far [24,25], while the transcription and translation initiation are the major rate-limiting steps for protein synthesis [24]. Through real-time quantitative PCR analysis, the enhanced transcript level of eGFP was revealed by fusing the signal peptide of *Bacillus thuringiensis* (Bt) Cry1Ia toxin (Iasp) to the N-terminal of eGFP [25]. However, the specific influencing mechanism remains to be further revealed. Verma et al. reported that the N-terminal amino acids encoded by codons 3 to 5 impact protein yield [24]. Moreover, the effect is independent of tRNA abundance, translation initiation efficiency, or overall mRNA structure [24]. According to this theory, we could boldly assume that some low efficient secretion SPs for one target protein identified through the detection of extracellular protein content may not only be attributed to their secreted properties but also limited by the N-terminal amino acids sequences’ gene transcription level or protein translation initiation. Likewise, the ideal signal peptide (SP) for a specific protein, ascertained through the identification of extracellular protein content, may also possess appropriate N-terminal nucleotide sequences to facilitate gene transcription or N-terminal amino acid sequences for efficient protein translation initiation.

Based on this hypothesis, following the identification of the optimal signal peptide (SP^-^LipA) out of five signal peptides from two distinct secretion pathways (Sec and TAT) by comparing the extracellular expression levels of an alkaline pectin lyase (APL) in *B. subtilis*, the initial five and seven amino acid sequences at the N-terminal were replaced to investigate their impacts on the expression and secretion of APL. A set of experimental designs has provided evidence that SP^-^LipA contributes to a higher extracellular expression of APL in *B. subtilis*. The increase in extracellular expression is not only attributed to the secretion ability of SP^-^LipA but also to the high transcription level of the SP^-^LipA-APL encoding gene due to the role of the initial N-terminal amino acid coding sequences. The highest extracellular APL activity produced by the engineered *B. subtilis* strain was up to 12,295 U/mL, which is the highest heterologous expression yield of alkaline pectinase in the prokaryotic expression system as far as those reported before.

## 2. Materials and Methods

### 2.1. Bacterial Strains, Plasmids and Chemicals

*E. coli* DH5 preserved in our laboratory was used for construction and storage of the recombinant plasmids. *Bacillus subtilis* SCK6, which served as the expression host, can be acquired from the Bacillus Genetic Stock Center (http://www.bgsc.org accessed on 24 May 2010) under the accession code 1A976 [26]. The mature peptide encoding sequence of a recombinant alkaline pectin lyase revealed in our previous report [27] was used as the target expression gene (*pelNK93I*) in *B. subtilis*. The pMA5 was used as a vector for heterologous expression of target genes in *B. subtilis* SCK6. The restriction endonucleases and DNA polymerase were commercially supplied by Thermo Fisher Scientific Co., Ltd., Waltham, MA, USA. All other chemicals and solutions with analytical reagent grade purity were purchased through commercial in China.

### 2.2. Screening of the Signal Peptides for Extracellular Production of APL

The encoding sequences of five signal peptides (Table 1) were inserted into the downstream of the promoter of pMA5 at the upstream of the mature peptide encoding sequence of APL, respectively. The recombinant plasmids were constructed using seamless cloning kit (GenScript Co., Ltd., Nanjing, China) according to manufacturer’s protocols [28]. After sequencing verification, the corrected recombinant plasmids were transformed into the competent cells of *B. subtilis* SCK6 according to the method described in our previous work [28]. The positive clones of the recombinant strains were cultured in SR medium (30 g/L tryptone, 50 g/L yeast extract, and 6 g/L K_2_HPO_4_) for shake flask cultivation at 37 °C and 220 rpm for 48 h. After cultivation, the culture was centrifugated at 12,000× *g* for 10 min. The supernatant was used for APL enzyme activity determination and protein electrophoresis.

### 2.3. Construction of the Recombinant Signal Peptides

Based on the original sequences of SP^-^LipA, SP^-^YncM, and SP^-^AmyX, substituted the encoding sequences of the N-terminal first 5 and 7 amino acids of SP^-^YncM and SP^-^AmyX by the corresponding sequences of SP^-^LipA, respectively. Meanwhile, the encoding sequences of the N-terminal first 5 and 7 amino acids of SP^-^YncM were also used to substitute the corresponding sequences of SP^-^LipA, respectively. All of the original and recombinant sequences (DNA and amino acids) of the signal peptides used in this work are listed in Table 1. The recombinant signal peptide encoding sequences were constructed by overlap-PCR using the primers in Appendix A and inserted into pMA5 by Megaprimer PCR to obtain the recombinant plasmids according to our previous report [29]. The recombinant plasmids were transformed into the competent cells of *B. subtilis* SCK6 and detected the extracellular expression of APL as described in Section 2.2. Here, the APL activity and protein expression in intracellular and inclusion bodies were also determined. The cells were treated with ultrasonication at 50 Hz for 10 min in an ice water bath to extract intracellular products. During the shake flask fermentation process, sample testing was performed at regular intervals. The secretion ratio was calculated as the proportion of extracellular APL activity to the sum of intracellular and extracellular APL activity. The biomass was displayed by measuring the dry cell weight of 1 mL culture dried for 24 h after centrifugation at 12,000× *g* for 10 min.

### 2.4. Enzyme Assay and SDS-PAGE

The APL activity was detected in our previous report [27]. One unit was defined as enzyme amount producing 1 μmol unsaturated polygalacturonic acid per minute from 0.2% pectin substrate at pH 9.0 and 60 °C for 10 min. The product yield was detected by employing an ELISA assay (BioTek, Epoch2TC, Agilent Technologies Co., Ltd., Santa Clara, California, USA) utilizing the extinction coefficient of 4600.0 L/(mol cm).

The target protein expression in extracellular, intracellular, and inclusion bodies was measured using sodium dodecyl sulfate polyacrylamide gel electrophoresis (SDS-PAGE) method as reported previously [9,28]. The electrophoresis was running in 5% stacking gel and 12% separating gel successively. The protein samples were dissolved in 4 × loading buffler and loaded 10 μL of the mixed solution in the loading hole. Coomassie Brilliant Blue G250 was used for staining the proteins.

### 2.5. Transcription Detection by Quantitative Real-Time PCR (qRT-PCR)

Appropriate amount of culture solution was taken from each test point and centrifuged at 12,000× *g* at 4 °C for 2 min to collect strain cells. The precipitated strain cells were resuspended using 100 μL TE solution (containing 3 mg/mL of lysozyme) and incubated at room temperature for 10 min. RNA isolation procedure and gene expression quantification were performed as previous reports [30,31]. The total RNA was extracted using a Simply P Total RNA Extraction Kit (BioFlux, Beijing, China). The quantitation was performed using the Nanodrop 2000 spectrophotometer (Thermo Fisher Scientific Fisher Scientific Co., Ltd., Waltham, MA, USA). The DNA in the mRNA extract samples was removed by DNase I [30]. The cDNA was synthesized using the PrimeScript RT reagent Kit (TaKaRa, Kusatsu, Japan) [30]. Primers used in qRT-PCR analysis to determine the expression levels of DanN (in the genome of *B. subtilis* SCK6) and APL (the recombinant plasmids) were listed in Appendix A. The relative expression level of APL was normalized to the DanN gene.

### 2.6. Fed-Batch Fermentation of the Engineered Strain to Produce Extracellular APL

Four hundred microlitres of original seed strains in LB medium (frozen at −80 °C) were inoculated into 400 mL fresh LB medium (Kan) in a 1 L shake flask and cultured at 220 rpm at 37 °C for 12 h. The cultured seed solution of 300 mL was inoculated into a 5 L fermenter containing 3 L fermentation medium for fermentation. Fermentation conditions were at constant temperature of 37 °C, pH 7.0 (pH adjusted with ammonia and lactic acid), aeration of 400 L/h, initial speed of 400 rpm (the value (30%) of dissolved oxygen (DO) was set as the threshold point for speed adjustment, 400–850 rpm). The process of feeding commences as soon as the levels of dissolved oxygen start to increase. The dry weight of strain cells and APL activity in fermentation liquid were measured every 6 h. The fermentation ends when both the dry weight and APL activity of the bacteria no longer rise. The fermentation medium contained sucrose (10 g/L), yeast extract powder (30 g/L), NaCl (8 g/L), KH_2_PO_4_ (3 g/L), MgCl_2_ (1 g/L), CaCl_2_·2H_2_O (0.3 g/L), MnSO_4_·2H_2_O (0.2 g/L), and ZnSO_4_·7H_2_O (0.02 g/L), and FeSO_4_·7H_2_O (0.02 g/L). The fed-batch feeding solution contained 400 g/L sucrose and 100 g/L yeast extract powder. Biomass refers to the dry weight of cells, which is determined by weighing the bacterial deposit after centrifuging 1 mL of bacterial solution and drying it at 80 °C for 24 h.

## 3. Results

### 3.1. Effects of Different Signal Peptides on the Secretory Expression of APL in B. subtilis

To screen the optimum signal peptide for secretory expression of the alkaline pectin lyase APL in *B. subtilis*, four commonly used signal peptides SP^-^LipA, SP^-^WapA, SP^-^AmyX, and SP^-^YncM were selected to guide the secretion of APL comparing with a pectinase signal peptide PelB. The results showed that the signal peptide SP^-^YncM belonging to the Sec pathway made a little lower extracellular activity of APL than that of SP^-^PelB after 48 h of fermentation (Figure 1A). The signal peptides SP^-^LipA and SP^-^WapA, which belonged to the TAT pathway, made 4.8 and 3.7-times higher extracellular activity of APL than that of SP^-^PelB, respectively (Figure 1A). However, the signal peptide SP^-^AymX made almost no APL expression in *B. subtilis* (Figure 1A,B). The dry cell weights of all the test strains showed similar values (Figure 1A). The extracellular target protein yields of the recombinant strains exhibited consistent relative yield trends with their extracellular activities following 48 h of fermentation (Figure 1A,B). As shown in Figure 1, SP^-^LipA was the optimum signal peptide for the secretory expression of APL in *B. subtilis*.

### 3.2. Effects of the N-Terminal 5–7 Amino acid Sequences of the Signal Peptides on the Secretory Expression of APL in B. subtilis

Based on the above results, the N-terminal 5 or 7 amino acid sequences of the optimum signal peptide SP^-^LipA were used to substitute the N-terminal amino acid sequences of SP^-^AmyX and SP^-^YncM, respectively. At the same time, the N-terminal 5 or 7 amino acid sequences of SP^-^YncM were used to substitute the N-terminal amino acid sequences of SP^-^LipA. The nucleotide and amino acid sequences of the recombinant signal peptides are shown in Table 1. As the results show in Figure 2A, when the N-terminal 5 and 7 amino acid sequences of SP^-^YncM were substituted with those of SP^-^LipA, both extracellular and intracellular activities of APL were slightly increased compared to those of the wild-type SP^-^YncM. However, when the N-terminal 5 and 7 amino acid sequences of SP^-^AmyX were substituted with those of SP^-^LipA, the extracellular activities of APL were enhanced by 13.2 and 18.4 times, and the intracellular activities of APL were also improved remarkably by 45.7 and 36.7 times, respectively (Figure 2A). Meanwhile, the results of the SDS-PAGE showed that the recombinant signal peptides SP^-^AmyX (SP^-^LipA_-_N5) and SP^-^AmyX (SP^-^LipA_-_N7) made higher expression levels of APL in extracellular, intracellular and inclusion bodies, compared with those guided by SP^-^AmyX, especially in intracellular (Figure 2B–D). However, when the N-terminal 5 and 7 amino acid sequences of SP^-^LipA were substituted with those of SP^-^YncM, both the extracellular and intracellular activities of APL decreased obviously (Figure 2A). The extracellular protein yields of APL were also decreased at a similar scale to the enzyme activities (Figure 2A,B). Moreover, any of the substitute N-terminal amino acid sequences could not influence the secretion ratio on a remarkable scale compared to their original signal peptides (Figure 2A).

The recombinant signal peptides SP^-^YncM (SP^-^LipA_-_N5) and SP^-^AmyX (SP^-^LipA_-_N5) were selected to detect the expression level both in extracellular and intracellular in the fermentation processing during 36 h with SP^-^LipA as a control. The recombinant signal peptide SP^-^YncM (SP^-^LipA_-_N5) gradually increased the extracellular activity of APL, while the intracellular activity remained at a steady low level (Figure 3A). However, using SP^-^AmyX (SP^-^LipA_-_N5) as the signal peptide, the intracellular activity of APL gradually increased with nearly no extracellular activity detected before 18 h of fermentation. After 24 h of fermentation, both extracellular and intracellular activity were gradually increased, while the intracellular activity was kept higher than the extracellular activity during the whole fermentation process (Figure 3A). Moreover, the growth state of the three strains did not show significant differences in the whole fermentation process (Figure 3B).

The N-terminal 5 amino acids of SP^-^LipA were used to directly attach to the N-terminal of the mature peptide of APL. It showed similar effects both on the extracellular and intracellular activities with those of the recombinant signal peptides SP^-^AmyX (SP^-^LipA_-_N5) during 48 h of fermentation (Figure 3A and Figure 4A). Similarly, the growths of the strains were not affected (Figure 4B).

### 3.3. Effects of the N-Terminal 5 Amino Acid Sequences of the Signal Peptides on the Gene Transcript in B. subtilis

The recombinant signal peptides SP^-^LipA (SP^-^YncM_-_N5), SP^-^YncM (SP^-^LipA_-_N5), and SP^-^AmyX (SP^-^LipA_-_N5) were employed for the comparison with their respective native signal peptides SP^-^LipA, SP^-^YncM, and SP^-^AmyX in relation to the gene transcript and protein expression of APL in *B. subtilis*. Meanwhile, SP^-^LipA_-_N5-APL, which was added to the sequences of SP^-^LipA_-_N5 directly to the N-terminal of the mature peptide of APL, was also employed for comparison with its mature peptide (APL) in relation to its gene transcript and protein expression in *B. subtilis*. During 48 h of fermentation, SP^-^LipA (SP^-^YncM_-_N5) resulted in a significant reduction in the total activities of APL compared to SP^-^LipA, while maintaining similar total activities as guided by SP^-^YncM (Figure 5A). The total activities of APL guided by SP^-^YncM (SP^-^LipA_-_N5) were higher than those guided by its original signal peptide SP^-^YncM with almost no change in the secretion ratio (Figure 5A). Moreover, both SP^-^AmyX (SP^-^LipA_-_N5) and SP^-^LipA_-_N5-APL could make remarkably higher total expression activities of APL than SP^-^AmyX-APL and APL, respectively (Figure 5A). The total expression activities of APL guided by SP^-^AmyX (SP^-^LipA_-_N5) or SP^-^ SP^-^LipA_-_N5-APL were even higher than those guided by SP^-^LipA (Figure 5A). Meanwhile, the growth of the eight strains at each time of the whole fermentation process showed no obvious difference (Figure 5B). On this basis, the relative qualification of the transcript levels of each strain at different fermentation times was determined by qRT-PCR. At the early expression time of 6 h, all of the recombinant strains showed the highest transcript level during the whole expression process, except the engineered strains containing SP^-^LipA_-_N5-APL and SP^-^AmyX-APL, respectively (Figure 5C). Moreover, the engineered strain containing signal peptide SP^-^AmyX showed a remarkably lower transcript level than others at 6 h fermentation and kept a relatively lower transcript level during the whole expression process (Figure 5C). The engineered strains containing signal peptide SP^-^YncM or APL showed the highest transcript level at 6 h of fermentation, but after that, the transcript level rapidly declined and kept to a relatively lower transcript level during the whole expression process (Figure 5C). It is important to highlight that the strains with the recombinant sequences SP^-^AmyX (SP^-^LipA_-_N5)-APL and SP^-^LipA_-_N5-APL and the native signal peptide SP^-^LipA-APL maintained significantly higher transcript levels throughout the majority of the expression process when compared to their counterparts (Figure 5C).

### 3.4. High Yield of APL through Fermentation of the Recombinant Strain in a 5-L Reactor

The engineered strain containing signal peptide SP^-^LipA, which showed the highest extracellular expression of APL, was used to perform a scale-up fermentation in a 5-L reactor. During 72 h of fermentation, the engineered strain produced the highest extracellular activity of APL of up to 12,295 U/mL at 60 h of fermentation (Figure 6), which was a 7.4-times increase from that produced by shake flask fermentation. The highest dry cell weight (47.2 g/L) was also obtained at 60 h of fermentation (Figure 6). The extracellular APL productivity of the engineered strain guided by signal peptide SP^-^LipA was 2.05 × 10^5^ U/L/h.

## 4. Discussion

*B. subtilis* is generally considered a good prokaryotic expression host for heterologous proteins based on its advances in extracellular protein secretion. Over the past decades, in order to make the efficient secretory expression of foreign proteins at a large scale in *B. subtilis*, more and more signal peptides were explored to improve the extracellular expression of different proteins. In this study, five signal peptides that had been verified to have an efficient secretion rate were detected to guide the extracellular expression of APL in *B. subtilis*. Among them, the signal peptides SP^-^YncM and SP^-^PelB belong to the typical Sec pathway, while SP^-^LipA, SP^-^WapA, and SP^-^AmyX belong to the TAT pathway with the conserved twin-arginine motif at the N-terminal of their sequences (Table 1). Based on comparing the extracellular APL activities of the recombinant strains connecting different signal peptides at the N-terminal of the mature peptide of APL, the TAT-type signal peptides SP^-^LipA and SP^-^WapA showed remarkably higher secretory expression than the Sec-type signal peptides SP^-^YncM and SP^-^PelB (Figure 1). While another TAT-type signal peptide SP^-^AmyX made obviously low expression both in and out of cells (Figure 1). Thus, it could not simply estimate any secretion pathway or signal peptide type as more suitable for the secretory expression of APL in *B. subtilis*. Meanwhile, because the signal peptides were on the N-terminal of the target proteins, the beginning of protein translation of the target proteins would also be related to the front sequence properties of different signal peptides, as it is well known that the initiation of translation is a major rate-limiting step for protein synthesis [24]. Therefore, the N-terminal signal peptides theoretically may not only determine the secretion rate but also play critical roles in the expression level of the target proteins. However, limited research has been conducted thus far on the impact of N-terminal signal peptides on the expression levels of heterologous genes.

Verma et al. conducted a study where they utilized a library of over 250,000 reporters along with in vitro and in vivo protein expression assays to examine the impact of early elongation on protein synthesis. They found that the specific amino acids encoded by codons 3 to 5 have an influence on protein yield [24]. Inspired by that study, we decided to selectively switch the N-terminal first five amino acid sequences of different signal peptides to assess their influences both on the expression and secretion of the target protein APL (Table 1). Moreover, we additionally designed substitutions for the first seven amino acid sequences to evaluate the effects of the typical twin-arginine motif on protein secretion (Table 1). When the N-terminal 5 and 7 amino acids of SP^-^LipA were substituted with those of SP^-^YncM, respectively, the extracellular activities of APL declined remarkably (Figure 2A). Interestingly, the recombinant signal peptide SP^-^LipA (SP^-^YncM_-_N7), which lacked the typical twin-arginine motif, showed a similar secretion ratio to the recombinant signal peptide SP^-^LipA (SP^-^YncM_-_N5), which retained the motif (Figure 2A,B). It indicated that the specific N-terminal amino acids sequence is the critical factor for the extracellular expression of the target protein rather than the secretion type. Meanwhile, the N-terminal 5 and 7 amino acids of SP^-^YncM and SP^-^AmyX were substituted by those of SP^-^LipA, leading to an increase in the extracellular activities of APL (Figure 2A). This indicates that the N-terminal sequence of SP^-^LipA has even greater advantages than those of SP^-^YncM and SP^-^AmyX in the synthesis of APL in *B. subtilis*. However, it is important to note that the N-terminal 5 and 7 amino acids sequence was not the only factor affecting the expression of the target proteins. When combined with the remnant sequences of SP^-^YncM and SP^-^AmyX, this led to a significantly different effect on the gene expression of APL (Figure 2). The recombinant signal peptides SP^-^AmyX (SP^-^LipA_-_N5) and SP^-^AmyX (SP^-^LipA_-_N7) exhibited notably high levels of APL expression, surpassing that guided by SP^-^LipA (Figure 2). This intriguing finding could pave the way for a new approach to enhancing the synthesis of diverse heterologous proteins in *B. subtilis*. However, the present study’s findings also suggested that the N-terminal amino acid sequence of the signal peptides had minimal impact on the extracellular secretion ratio of the target proteins (Figure 2A).

As shown in Figure 2A, SP^-^YncM (SP^-^LipA_-_N5) and SP^-^AmyX (SP^-^LipA_-_N5) had relatively more significant influences in the expression level of APL than SP^-^YncM (SP^-^LipA_-_N7) and SP^-^AmyX (SP^-^LipA_-_N7), respectively. Thus, the engineered strains containing the recombinant signal peptide SP^-^YncM (SP^-^LipA_-_N5) and SP^-^AmyX (SP^-^LipA_-_N5) were chosen to further evaluate the APL expression difference during the whole fermentation process. As shown in Figure 3, SP^-^LipA and SP^-^YncM (SP^-^LipA_-_N5) mainly produced extracellular APL activities even at the beginning fermentation phase at 6 h, while SP^-^AmyX (SP^-^LipA_-_N5) mainly produced intracellular APL activities during the whole fermentation process and the extracellular activity began increasing after 18 h of fermentation. Inferred from this result, SP^-^LipA_-_N5 just had the effect of improving the gene expression and it may destroy the typic twin-arginine motif thus leading to SP^-^AmyX (SP^-^LipA_-_N5) losing its secretory function. As for a portion of the extracellular APL activity detected, it is likely caused by cell lysis in the late fermentation period. To test this hypothesis, SP^-^LipA_-_N5 was directly attached to the N-terminal of the mature peptide of APL. As shown in Figure 4, SP^-^LipA_-_N5 indeed remarkably enhanced the expression of APL. As the strain containing the mature peptide of APL detected extracellular APL activity after 12 h of fermentation, similarly the extracellular activity of the strain containing SP^-^LipA_-_N5-APL is also due to cell lysis after 12 h of fermentation (Figure 4). Thus, the hypothesis regarding the impact of SP^-^LipA_-_N5 on the expression of target proteins was validated.

To sum up the above results, the recombinant signal peptide SP^-^YncM (SP^-^LipA_-_N5) caused increased expression with no effect on its secretion, while the recombinant signal peptides SP^-^AmyX (SP^-^LipA_-_N5) caused remarkably increased expression with secretion function lost (Figure 5A). The engineered strains were analyzed for the relative quantification of transcript levels, revealing that the recombinant proteins containing the SP^-^LipA_-_N5 sequence at their N-terminal exhibited significantly higher transcript levels compared to the control strains (Figure 5C). Moreover, the high expression level strains also showed consistently high transcript levels during the fermentation process (Figure 5C). Thus, the specific N-terminal sequence SP^-^LipA_-_N5 was discovered to have the capability to enhance the production of foreign proteins in *B. subtilis*. Meanwhile, it is also suggested that researchers should also consider the effect of the N-terminal sequence on expression when selecting appropriate signal peptides in *B. subtilis*.

SP^-^LipA was identified as the optimum signal peptide for the heterologous extracellular expression of APL with both high expression levels based on its N-terminal specific sequence and efficient secretion rate through the TAT pathway in *B. subtilis*. Fed-batch culture in a 5-L reactor was performed to test the industrial fermentation capacity of the engineered strain. The highest extracellular APL activity (12,295 U/mL) was obtained at 60 h of fermentation, which further exceeded the total soluble APL activity (10,181 U/mL) expressed in the recombinant *Escherichia coli* strain we had constructed [27]. So far, it is the highest heterologous expression yield of alkaline pectinase in prokaryotic expression system as far as what has been reported before [27].

## 5. Conclusions

Collectively, our data underscore the critical role played by the first N-terminal 5–7 amino acids of different signal peptides in facilitating the expression of heterologous APL in *B. subtilis*. This study has unveiled, for the first time, that the first five amino acids at the N-terminus of the signal peptide SP^-^LipA can remarkably improve the expression of the target protein APL by enhanced transcriptional level in *B. subtilis*. This will open a new perspective of signal peptide screening for the expression of heterologous proteins. Moreover, this study has achieved a new milestone in the extracellular production of APL in *B. subtilis*, reaching a record-high yield of 12,295 U/mL.

## Figures and Tables

**Figure 1 microorganisms-12-01275-f001:**
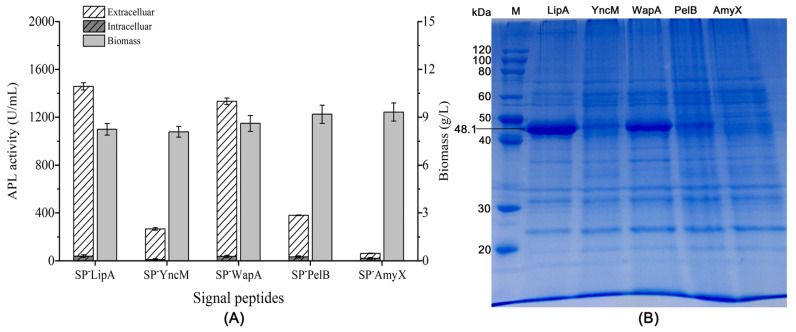
Effects of different signal peptides on the extracellular expression of APL: (**A**) APL activities in the extracellular and intracellular, respectively. All values are expressed as mean ± SD (*n* = 3); (**B**) extracellular proteins detected by SDS-PAGE.

**Figure 2 microorganisms-12-01275-f002:**
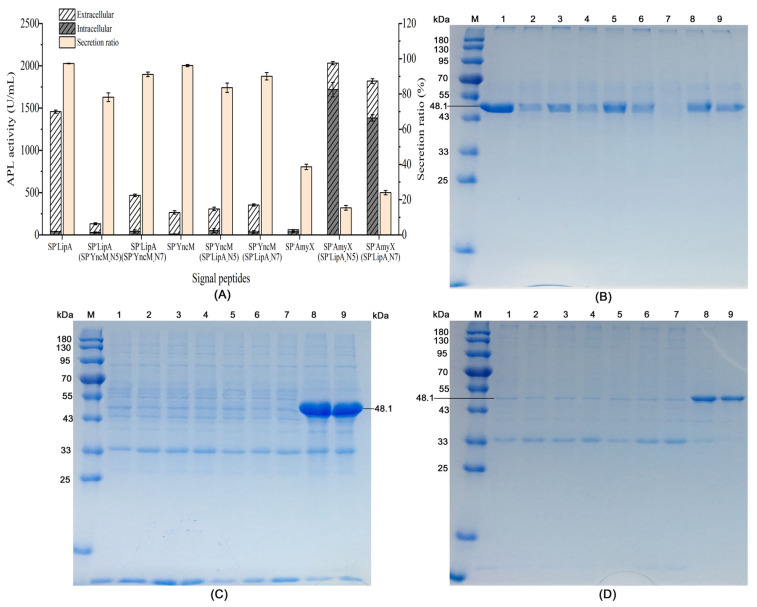
Effects of the recombinant signal peptides on the expression and secretion of APL: (**A**) Activities and secretion ratio of APL guided by different signal peptides. All values are expressed as mean ± SD (*n* = 3); (**B**) extracellular proteins detected by SDS-PAGE; (**C**) intracellular proteins detected by SDS-PAGE; (**D**) inclusion bodies detected by SDS-PAGE. M: protein molecular weight markers; lane 1–9 represented APL expression directed by distinct signal peptides. Lane 1: SP^-^LipA; Lane 2: SP^-^LipA (SP^-^YncM_-_N5); Lane 3: SP^-^LipA (SP^-^YncM_-_N7); Lane 4: SP^-^YncM; Lane 5: SP^-^YncM (SP^-^LipA_-_-N5); Lane 6: SP^-^YncM (SP^-^LipA_-_N7); Lane 7: SP^-^AmyX; Lane 8: SP^-^AmyX (SP^-^LipA_-_N5); Lane 9: SP^-^AmyX (SP^-^LipA_-_N7).

**Figure 3 microorganisms-12-01275-f003:**
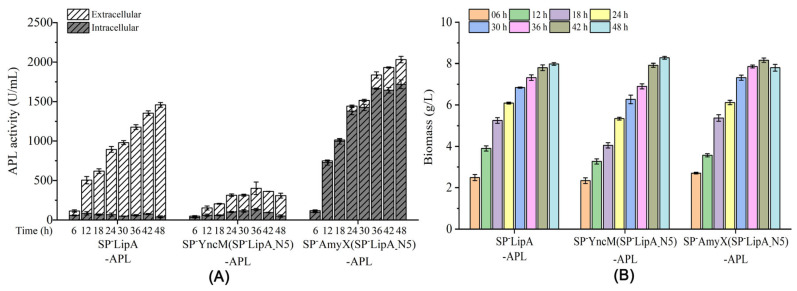
The APL activity profiles (**A**) and strain growth profiles (**B**) of the recombinant strains expressing APL directed by distinct signal peptides SP^-^LipA, SP^-^YncM (SP^-^LipA_-_N5), and SP^-^AmyX (SP^-^LipA_-_N5), respectively. All values are expressed as mean ± SD (*n* = 3).

**Figure 4 microorganisms-12-01275-f004:**
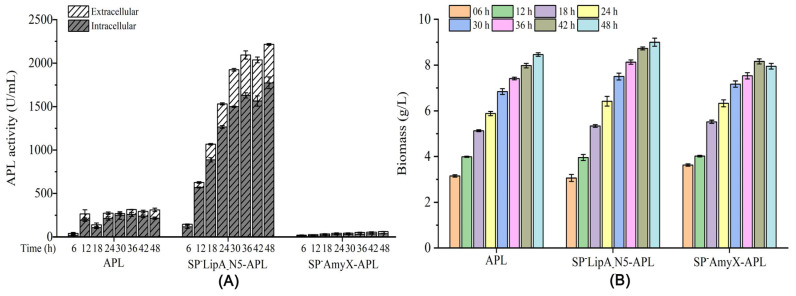
The APL activity profiles (**A**) and strain growth profiles (**B**) of the recombinant strains expressing APL, SP^-^LipA_-_N5-APL, and SP^-^AmyX-APL, respectively. All values are expressed as mean ± SD (*n* = 3).

**Figure 5 microorganisms-12-01275-f005:**
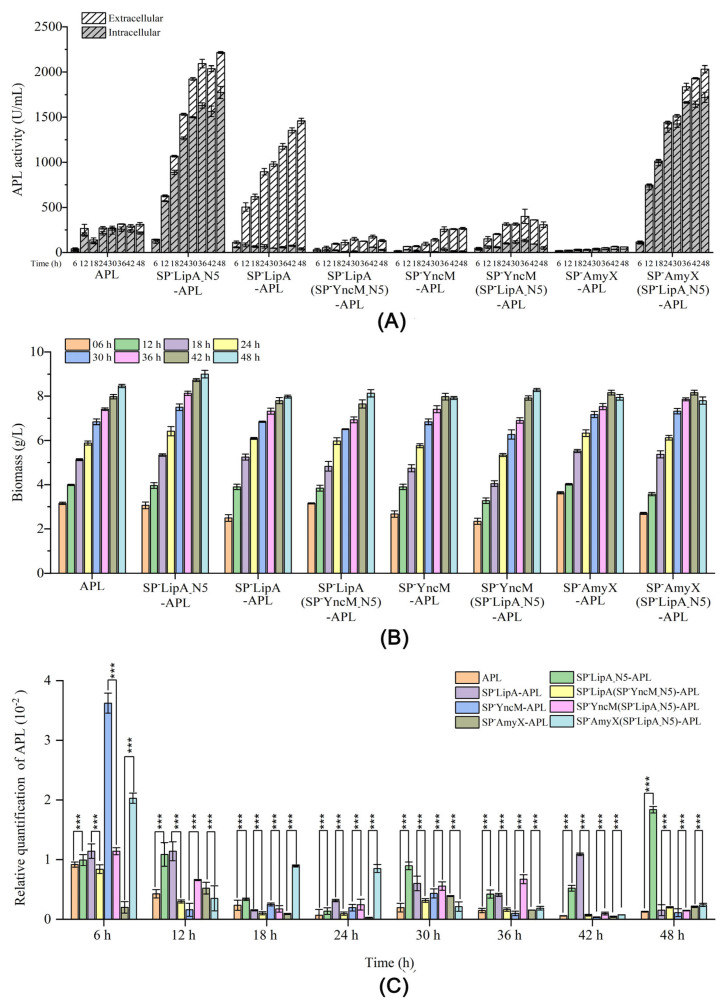
The APL activity profiles (**A**), strain growth profiles (**B**), and transcriptional level profiles (**C**) of all the recombinant strains constructed in this work. All values are expressed as mean ± SD (*n* = 3); Statistical analysis was carried out using GraphPad Prism 6.01 (GraphPad Software Inc., San Diego, CA, USA), “***” means *p* < 0.001.

**Figure 6 microorganisms-12-01275-f006:**
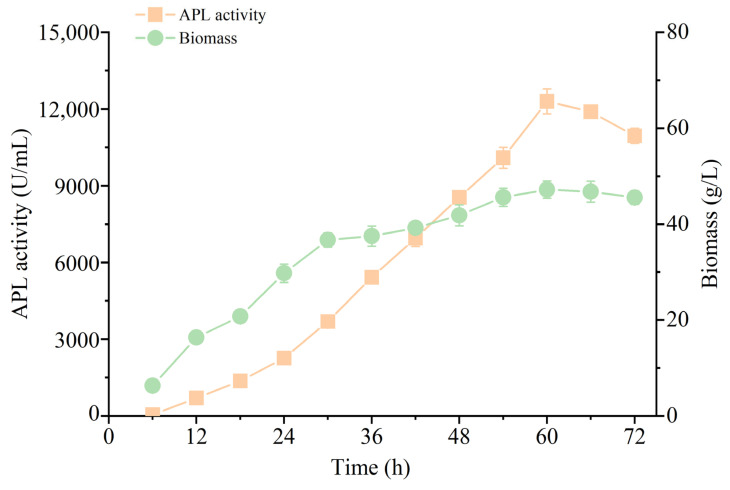
Scale-up fermentation in a 5 L reactor of the engineered strain *B. subtilis* SCK6 (pMA5-SP^-^LipA-APL). All values are expressed as mean ± SD (*n* = 3).

**Table 1 microorganisms-12-01275-t001:** Nucleotide and amino acid sequences of the signal peptides used in this work.

Signal Peptide	Nucleotide Sequence	Amino Acid Sequence
SP^-^LipA	ATGAAATTTGTGAAACGCAGAATTATTGCGCTGGTGACAATTCTGATGCTGAGCGTGACAAGCCTGTTTGCGCTGCAACCGAGCGCGAAAGCG	MKFVK**RR**IIALVTILMLSVTSLFALQPSAKA
SP^-^YncM	ATGGCTAAACCGCTGTCAAAAGGCGGCATTCTGGTTAAAAAAGTTCTGATTGCAGGCGCAGTTGGCACAGCAGTCCTGTTTGGCACGCTGAGTAGCGGCATTCCGGGACTGCCAGCAGCTGATGCG	MAKPLSKGGILVKKVLIAGAVGTAVLFGTLSSGIPGLPAADA
SP^-^WapA	ATGAAAAAACGCAAACGCAGAAATTTTAAACGCTTTATTGCGGCGTTTCTGGTTCTGGCGCTGATGATTAGCCTGGTTCCGGCGGATGTGCTGGCG	MKKRK**RR**NFKRFIAAFLVLALMISLVPADVLA
SP^-^PelB	ATGAAATACCTGCTGCCGACCGCTGCTGCTGGTCTGCTGCTCCTCGCTGCCCAGCCGGCGATGGCC	MKYLLPTAAAGLLLLAAQPAMA
SP^-^AmyX	ATGGTCAGCATCCGCCGCAGCTTCGAAGCGTATGTCGATGACATGAATATCATTACTGTTCTGATTCCTGCTGAACAAAAGGAAATCATGACACCGCCG	MVSI**RR**SFEAYVDDMNIITVLIPAEQKEIMTPP
SP^-^LipA(SP^-^YncM_-_N5)	**ATGGCTAAACCGCTG**CGCAGAATTATTGCGCTGGTGACAATTCTGATGCTGAGCGTGACAAGCCTGTTTGCGCTGCAACCGAGCGCGAAAGCG	MAKPL**RR**IIALVTILMLSVTSLFALQPSAKA
SP^-^LipA(SP^-^YncM_-_N7)	**ATGGCTAAACCGCTGTCAAAA**ATTATTGCGCTGGTGACAATTCTGATGCTGAGCGTGACAAGCCTGTTTGCGCTGCAACCGAGCGCGAAAGCG	MAKPLSKIIALVTILMLSVTSLFALQPSAKA
SP^-^YncM(SP^-^LipA_-_N5)	**ATGAAATTTGTGAAA**TCAAAAGGCGGCATTCTGGTTAAAAAAGTTCTGATTGCAGGCGCAGTTGGCACAGCAGTCCTGTTTGGCACGCTGAGTAGCGGCATTCCGGGACTGCCAGCAGCTGATGCG	MKFVKSKGGILVKKVLIAGAVGTAVLFGTLSSGIPGLPAADA
SP^-^YncM(SP^-^LipA_-_N7)	**ATGAAATTTGTGAAACGCAGA**GGCGGCATTCTGGTTAAAAAAGTTCTGATTGCAGGCGCAGTTGGCACAGCAGTCCTGTTTGGCACGCTGAGTAGCGGCATTCCGGGACTGCCAGCAGCTGATGCG	MKFVKRRGGILVKKVLIAGAVGTAVLFGTLSSGIPGLPAADA
SP^-^AmyX(SP^-^LipA_-_N5)	**ATGAAATTTGTGAAA**CGCAGCTTCGAAGCGTATGTCGATGACATGAATATCATTACTGTTCTGATTCCTGCTGAACAAAAGGAAATCATGACACCGCCG	MKFVKRSFEAYVDDMNIITVLIPAEQKEIMTPP
SP^-^AmyX(SP^-^LipA_-_N7)	**ATGAAATTTGTGAAACGCAGA**TTCGAAGCGTATGTCGATGACATGAATATCATTACTGTTCTGATTCCTGCTGAACAAAAGGAAATCATGACACCGCCG	MKFVKRRFEAYVDDMNIITVLIPAEQKEIMTPP

Note: The substitute sequences were marked using underline. The twin-arginine motif was represented as bold font.

## Data Availability

All data generated or analyzed during this study are included in this published article and its Appendix A. All further data will be provided by the corresponding author at any time upon request.

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
