# Peer review of "N-Terminal Sequences of Signal Peptides Assuming Critical Roles in Expression of Heterologous Proteins in Bacillus subtilis"

_microorganisms, 2024, doi:10.3390/microorganisms12071275_

Round 1

Reviewer 1 Report

Comments and Suggestions for Authors

The article „N-terminal sequences of signal peptides assuming critical roles in expression of heterologous proteins in Bacillus subtilis“ by Zhang et al. deals with testing of different N-terminal sequences to increase expression and secretion of proteins in B. subtilis.

In its present form, the paper is extremely difficult to read and understand, and I feel that the authors are mixing up several fundamental aspects in biology.

I will give examples line by line; in the end, I believe that work could only be published if radically modified.

20: Genes do not have N-termini, only proteins do

24: “signal peptide LipA” do not give protein names to signal sequences!!!

25 Meanwhile, it also revealed that the signal peptide LipA could lead to higher secretory expression than other signal peptides  not because of its sequence structure associated with secretion is stronger but because the N-terminal sequences showed greater advantages in gene transcription” This sentence is almost incomprehensible; again, protein is confused with gene

50 please explain the nature and composition of signal sequences. It appears that the authors are not aware that signal sequences are usually of a length of 20 or more amino acids, and much more complex than just the first 7 amino acids. Please also explain differences in SPs of the SRP and the Tat pathway – this extends further than just the twin arginines!

55 “the N-terminal SP sequence is also the initiation of transcription and translation of the target genes so far.” I am trying to remain professional, but this is really completely wrong. Transcription does not start with codons of the SP. Transcription starts upstream of the start codon and also upstream of the ribosome binding site. It is extremely rarely that mRNA starts with a start codon, vastly, there are several tens of bases upstream of the start codon (and thus of the start of the SP), such that sequences of the SP can hardly affect transcription. It is possibly that efficient translation has an effect on transcription, but this could go both ways, i.e. increased translation efficiency may turn down transcription.

61 sentence is not understandable

73 “persistently high transcriptional level of the N-terminal amino acids.” Transcription is confused with translation

161 It took me a while to understand that the authors have used the first few amino acids of proteins such as LipA to change the SP of APL.

“four commonly used signal peptides DERIVED FROM THE FIRST 5 or 7 AMINO ACIDS of LipA ….” Please explain in detail: were 5 or 7 aa added to the N-terminus of APL (please state the correct protein and gene name), or were the N-terminal amino acids of APL exchanged for other sequences?

As a consequence of this uncertainty: the authors measure activity of modified APL from the supernatant: is it clear that all modified SPs are cleaved off? And is it clear that all modified APL variants have a similar activity when purified?

174 “extracellular activity” is likely meant, because proteins are not expressed extracellularly

Table 1: please mark twin arginines.

How can the authors be sure that a non-TAT substrate protein is efficiently secreted by TAT. When only the five first aa are changed? TAT interacts with more than just the twin arginines.

Fig. 2 how is “secretion rate” determined?

253 except the engineered strain containing signal peptide APL (LipA N5) which showed the highest transcript level at 12 h fermentation (Figure 5 C).” what about the APL LipA N5 level after 48 h?

Fig. 5 please show a growth curve so it becomes apparent when cells are in exponential phase or in stationary phase

260 “However, the engineered strains containing the recombinant signal peptide (AmyX (LipA N5) and APL (LipA N5)) and the original signal peptide LipA kept relatively higher transcript level during the whole expression  process (Figure 5 C)” this is a central claim of the work, but I can not see any correlation between sequences and transcription (mRNA) levels. If the authors want to maintain this claim, they have to show this in a convincing manner, including statistical significances.

327 This sentence is hard to grasp, and it seems to contradict the results and the title. As I understand it, here the authors admit that there is no correlation between different sequences at the start of the SPs and transcript levels, and thus, this work does not show that SP sequences can affect intracellular expression rates. 

As a note to this: the authors ignore that even if transcription rates would be the same, a change in the SP sequence could affect secretion efficiency. Non-secreted (modified) APL may be rapidly degraded within the cell. Thus, different SPs may affect expression rates by increasing or decreasing secretion efficiency, thereby affecting intracellular protein proteolysis rates. I think the authors should make experiments testing intracellular protein levels dependent on different SP sequences, and not only extracellular activity.

Comments on the Quality of English Language

Many sentences are grammatically incorrect such that the meaning can only be guessed, or becomes more clear in later sentences.

Author Response

All changes are made with red font highlights in the revised manuscript.

Comments of reviewer 1:

The article “N-terminal sequences of signal peptides assuming critical roles in expression of heterologous proteins in Bacillus subtilis” by Zhang et al. deals with testing of different N-terminal sequences to increase expression and secretion of proteins in B. subtilis.

In its present form, the paper is extremely difficult to read and understand, and I feel that the authors are mixing up several fundamental aspects in biology.

I will give examples line by line; in the end, I believe that work could only be published if radically modified.

20: Genes do not have N-termini, only proteins do

Response: Thank the reviewer for the correction of the concept. For this question, we have corrected every misrepresentation in the revised manuscript (Line 20, 79-80, and 89……).

24: “signal peptide LipA” do not give protein names to signal sequences!!!

Response: The protein name of the signal sequences has added in Line 24 of the revised manuscript.

25 Meanwhile, it also revealed that the signal peptide LipA could lead to higher secretory expression than other signal peptides not because of its sequence structure associated with secretion is stronger but because the N-terminal sequences showed greater advantages in gene transcription” This sentence is almost incomprehensible; again, protein is confused with gene

Response: The sentence has been reedited to make it more comprehensible (line 25-27).

50 please explain the nature and composition of signal sequences. It appears that the authors are not aware that signal sequences are usually of a length of 20 or more amino acids, and much more complex than just the first 7 amino acids. Please also explain differences in SPs of the SRP and the Tat pathway – this extends further than just the twin arginines!

Response: The common knowledge about signal peptides has been added in the revised manuscript (line 52-62) according to the reviewer’s comment. Besides, we also like to explain more information about the effects of signal peptide sequences on gene transcription and protein translation which are more important to discuss in this paper and are also of concern to the reviewer (Line 68-71).

55 “the N-terminal SP sequence is also the initiation of transcription and translation of the target genes so far.” I am trying to remain professional, but this is really completely wrong. Transcription does not start with codons of the SP. Transcription starts upstream of the start codon and also upstream of the ribosome binding site. It is extremely rarely that mRNA starts with a start codon, vastly, there are several tens of bases upstream of the start codon (and thus of the start of the SP), such that sequences of the SP can hardly affect transcription. It is possibly that efficient translation has an effect on transcription, but this could go both ways, i.e. increased translation efficiency may turn down transcription.

Response: Thanks very much for the correction of the concept of transcription initiation. We actually know the existence of 5’UTR and the transcript initiation site is TSS but not ATG. However, the sequences of SP indeed make positive influence on the transcript level of its downstream genes which had been revealed in the previous report (Gao et al. Microb Cell Fact (2020) 19:112). But, so far, the specific influence mechanism remains not clear which is the key research significance of this work. In order to make more accurate expression, we have substituted the “initiation of transcription” to “initial extension of genes transcription”. Meanwhile, all of the misrepresentation like this in the full text have been corrected in the revised manuscript. (Line 66-67,77,80,318)

61 sentence is not understandable

 Response: The sentence has been reedited in the revised manuscript (Line 74-77).

73 “persistently high transcriptional level of the N-terminal amino acids.” Transcription is confused with translation

 Response: It has been corrected in the revised manuscript (Line 87-89).

161x It took me a while to understand that the authors have used the first few amino acids of proteins such as LipA to change the SP of APL.rr

“four commonly used signal peptides DERIVED FROM THE FIRST 5 or 7 AMINO ACIDS of LipA ….” Please explain in detail: were 5 or 7 aa added to the N-terminus of APL (please state the correct protein and gene name), or were the N-terminal amino acids of APL exchanged for other sequences?

Response: Here, we were confused by the reviewer referring sentence “four commonly used signal peptides DERIVED FROM THE FIRST 5 or 7 AMINO ACIDS of LipA ….” which could not find in the text. In the line 161 of the original manuscript the sentence is “To screen the optimum signal peptide for secretory expression of the alkaline pectin lyase APL in B. subtilis, four commonly used signal peptides LipA, WapA, AmyX, and YncM were selected to guide the secretion of APL comparing with a pectinase signal peptide PelB.” We believe that the expression is clear enough to understand. As the correct protein name of APL, we have already given in the Abstract section when referred in the text for the first time and we did not use the gene name of APL in the text. We guess the key point of your concern may be answered as follow station. The N-terminal amino acids of the mature peptides of APL were not been exchanged for other sequences. The detailed information of sequence exchanges please find in section 2.3 and Table 1.

As a consequence of this uncertainty: the authors measure activity of modified APL from the supernatant: is it clear that all modified SPs are cleaved off? And is it clear that all modified APL variants have a similar activity when purified?

Response: In this work, except APLLipA N5, there is no modified APL, detection of the extracellular APL means that the modified SPs have been cleaved off. However, the intracellular APL with the modified SPs also showing the corresponding activity with the protein expression (Figure 2) which indicated that the signal peptide sequences made little influence to the activity of APL.

174 “extracellular activity” is likely meant, because proteins are not expressed extracellularly

Response: As shown in Figure 1B, most of the proteins were expressed extracellularly. So, we believe that the original title of Figure 1 is more accurate which contains all the meanings of A and B.

Table 1: please mark twin arginines.

Response: The twin arginines have been marked with bold in Table 1.

How can the authors be sure that a non-TAT substrate protein is efficiently secreted by TAT. When only the five first aa are changed? TAT interacts with more than just the twin arginines.

Response: In this work, we don’t judge if the non-TAT substrate protein is efficiently secreted by TAT basing on supposes, but all of the results were showed both extracellular and intracellular activities led by modified SPs.

Fig. 2 how is “secretion rate” determined?

Response: The expression is not accurate enough. We have modified it as “Secretion ratio” which is determined by the ratio of extracellular activity compared to total activity.

253 except the engineered strain containing signal peptide APL (LipA N5) which showed the highest transcript level at 12 h fermentation (Figure 5 C).” what about the APL LipA N5 level after 48 h?

Response: Because it showed no variation trend of transcript level with the change of time in Figure 5C, the transcript level after 48 h is hard to predict. Thus, only the comprehensive transcription level of each variant in the whole fermentation process and its actual protein expression level can be evaluated to judge the effect of gene transcription on the final protein expression.

Fig. 5 please show a growth curve so it becomes apparent when cells are in exponential phase or in stationary phase

Response: Actually, Figure 5B could represent the growth curve of each strain. Through the results of Figure 5B, all of the strains showed similar profile in growth. Besides, the relative quantification of transcript level of each variant showed in Figure 5C had already excluded the effect of different growth periods.

260 “However, the engineered strains containing the recombinant signal peptide (AmyX (LipA N5) and APL (LipA N5)) and the original signal peptide LipA kept relatively higher transcript level during the whole expression process (Figure 5 C)” this is a central claim of the work, but I can not see any correlation between sequences and transcription (mRNA) levels. If the authors want to maintain this claim, they have to show this in a convincing manner, including statistical significances.

Response: All of the results of Figure 5 have been rearranged to be easier to analyze. Besides, the statistical analysis results regarding the significance of the pairwise comparison of data in Figure 5C have been included.

327 This sentence is hard to grasp, and it seems to contradict the results and the title. As I understand it, here the authors admit that there is no correlation between different sequences at the start of the SPs and transcript levels, and thus, this work does not show that SP sequences can affect intracellular expression rates. 

Response: We did not admit that there is no correlation between different sequences at the start of the SPs and transcript levels in any part of the paper. In the part that the reviewer referred to, we just inferred that the N-terminal amino acids sequence of the signal peptides did not make significant effect on the extracellular secretion ratio of the target proteins. It did not contradict the results and the title. We have reedited that part to be more understandable (Line 347-353).

As a note to this: the authors ignore that even if transcription rates would be the same, a change in the SP sequence could affect secretion efficiency. Non-secreted (modified) APL may be rapidly degraded within the cell. Thus, different SPs may affect expression rates by increasing or decreasing secretion efficiency, thereby affecting intracellular protein proteolysis rates. I think the authors should make experiments testing intracellular protein levels dependent on different SP sequences, and not only extracellular activity.

Response: In this study, all of the results were showed both extracellular and intracellular activities led by modified SPs. Please see Figure1-Figure 5. The potential for rapid intracellular protein degradation, as suggested by reviewers, cannot be accurately determined or measured by assessing intracellular protein expression levels. Our study focused on analyzing changes in gene transcription levels prior to the occurrence of these issues and identifying patterns. 

Reviewer 2 Report

Comments and Suggestions for Authors

This manuscript presents an interesting study on the effects of different signal peptides' N-terminal 5-7 amino acid sequences on gene transcription and protein secretion, like the alkaline pectin lyase (APL) in the Bacillus subtilis strain. The authors found that the N-terminal sequences showed more significant advantages in gene transcription, resulting in the highest extracellular yields (12295 U/mL) of APL in B. subtilis. These results significantly contribute to the area, confirming this study's publication potential. Some minor suggestions are presented below.

Comments to authors

Materials and methods

-Lines 79, 84, 104: review the writing of some words.

-Lines 97, 117, 131: 12000 x g (how the centrifugation speed is presented).

-Line 143: In which medium were the original seed strains stored? Include this information.

-Line 144: Did the inoculation take place in a shaken flask? Include this information.

-Line 145: Why was an inoculum concentration greater than 10% (v/v) used?

-Line 147: Replace "ventilation" with "aeration".

-Line 148: Present the meaning of DO (dissolved oxygen).

-Lines 152-154: Was the culture medium based on any previous study? If yes, please cite it.

-Line 155: Why was the C/N ratio not maintained in the feed, compared to the initial medium without feed?

-Line 155: Was the cell biomass measurement made for each sample collected or using a standard curve?

Results

-Line 163-164: Review whether the information is correct based on Figure 1 A: "... YncM belonging to the Sec pathway Made a little higher extracellular activity of APL than that of PelB".

-Line 185: Review whether the increase order is correct based on Figure 2 A: "…were increased by 3.7 and 1.6 times, respectively."

-Line 217: Review whether the information is correct based on Figure 3 A: "...12 h fermentation. After 15 h fermentation..."

-Line 254: Review whether the information is correct based on Figure 5 C: "...which showed the highest transcript level at 12 h fermentation...".

-Line 275-276: Review whether the information is correct based on Figure 6: "...The highest wet cell weight (188 g/L) was also obtained at 60 h fermentation".

Discussion

-Review the items indicated above in the discussion of the results.

Comments on the Quality of English Language

Minor editing of English language required

Author Response

Comments of reviewer 2:

This manuscript presents an interesting study on the effects of different signal peptides' N-terminal 5-7 amino acid sequences on gene transcription and protein secretion, like the alkaline pectin lyase (APL) in the Bacillus subtilis strain. The authors found that the N-terminal sequences showed more significant advantages in gene transcription, resulting in the highest extracellular yields (12295 U/mL) of APL in B. subtilis. These results significantly contribute to the area, confirming this study's publication potential. Some minor suggestions are presented below.

Comments to authors

Materials and methods

-Lines 79, 84, 104: review the writing of some words.

Response: The words have been corrected in the revised manuscript (Line 95,100,120).

-Lines 97, 117, 131: 12000 x g (how the centrifugation speed is presented).

Response: All of the centrifugation speed units have been corrected in the revised manuscript (Line 114,133,148).

-Line 143: In which medium were the original seed strains stored? Include this information.

Response: The medium of the original seed strains was added in the revised manuscript (Line 160).

-Line 144: Did the inoculation take place in a shaken flask? Include this information.

Response: Yes, the information has been added in the revised manuscript (Line 161).

-Line 145: Why was an inoculum concentration greater than 10% (v/v) used?

Response: Actually, the inoculum concentration is 10%. It has been corrected in the revised manuscript (Line 162).

-Line 147: Replace "ventilation" with "aeration".

Response: It has been corrected in the revised manuscript (Line 161).

-Line 148: Present the meaning of DO (dissolved oxygen).

Response: It has been presented in the revised manuscript (Line 165).

-Lines 152-154: Was the culture medium based on any previous study? If yes, please cite it.

Response: No, it was optimized in our lab.

-Line 155: Why was the C/N ratio not maintained in the feed, compared to the initial medium without feed?

Response: It mainly based on the fermentation optimization results for the engineered B. subtilis strains in our lab which hasn't been published before.

-Line 155: Was the cell biomass measurement made for each sample collected or using a standard curve?

Response: It was made for each sample collected.

 Results

-Line 163-164: Review whether the information is correct based on Figure 1 A: "... YncM belonging to the Sec pathway Made a little higher extracellular activity of APL than that of PelB".

Response: It has been corrected in the revised manuscript (Line 182).

-Line 185: Review whether the increase order is correct based on Figure 2 A: "…were increased by 3.7 and 1.6 times, respectively."

Response: It has been corrected in the revised manuscript (Line 202-208).

-Line 217: Review whether the information is correct based on Figure 3 A: "...12 h fermentation. After 15 h fermentation..."

Response: It has been corrected in the revised manuscript (Line 236).

-Line 254: Review whether the information is correct based on Figure 5 C: "...which showed the highest transcript level at 12 h fermentation...".

Response: It has been corrected in the revised manuscript (Line 274-275).

-Line 275-276: Review whether the information is correct based on Figure 6: "...The highest wet cell weight (188 g/L) was also obtained at 60 h fermentation".

 Response: It has been corrected in the revised manuscript (Line 295-296).

Discussion

- Review the items indicated above in the discussion of the results.

Response: The whole discussion part has been checked and corrected carefully.

Unfortunately, this manuscript needs extensive editing for English expression, spelling and grammar.  Having a native English speaker edit it is probably essential.

Response: A carefully English editing of the whole manuscript has been made by a native English-speaking colleague.

Reviewer 3 Report

Comments and Suggestions for Authors

Unfortunately, this manuscript needs extensive editing for English expression, spelling and grammar.  Having a native English speaker edit it is probably essential.

The manuscript describes research into the effect of the N-terminal signal peptide on the expression yield of alkaline pectin lyase.  The underlying theory is that the N-terminal amino acids encoded by codons 3 to 5 of the signal peptide impact protein yield.  The authors admit that this is not the only factor determining expression and protein yield, but it is a good hypothesis and the research being presented provides good evidence for it. 

The study reported here is a clear demonstration of this hypothesis, well carried out, and with accurate data to support the hypothesis.

The major problem here is the difficulty of understanding all of the data and procedures, because of the lack of adequate and accurate English.   

Comments on the Quality of English Language

see above

Author Response

Comments of reviewer 3:

The manuscript describes research into the effect of the N-terminal signal peptide on the expression yield of alkaline pectin lyase.  The underlying theory is that the N-terminal amino acids encoded by codons 3 to 5 of the signal peptide impact protein yield.  The authors admit that this is not the only factor determining expression and protein yield, but it is a good hypothesis and the research being presented provides good evidence for it. 

The study reported here is a clear demonstration of this hypothesis, well carried out, and with accurate data to support the hypothesis.

The major problem here is the difficulty of understanding all of the data and procedures, because of the lack of adequate and accurate English.   

Response: A carefully English editing of the whole manuscript has been made by a native English-speaking colleague.

Round 2

Reviewer 1 Report

Comments and Suggestions for Authors

The manuscript is much improved. Fig. 5 has 3 panels, all of which are a lot too small. Otherwise I am satisfied with the changes done.

Author Response

Commont 1: The manuscript is much improved. Fig. 5 has 3 panels, all of which are a lot too small. Otherwise I am satisfied with the changes done.

Response: Many thanks for the reviewer’s valuable comments, which has greatly contributed to improving the academic quality of this work. The Figure 5 has been improved and resubmitted.